# Layer-based 3D Gaussian Splatting for Sparse-view CT Reconstruction

## Abstract

We introduce a dynamic framework for 3D sparse-view Gaussian Splatting that learns scene representations through layerwise, iterative refinement of the Gaussian primitives. Conventional methods typically rely on dense, one-time initialization, where the placement of Gaussians is guided by 2D projection supervision and density control. However, such strategies can lead to misalignment with the true 3D structure, particularly in regions with insufficient projection information due to sparse-view acquisition. In contrast, we adopt a coarse-to-fine approach beginning with a base representation and progressively expanding it by adding new layers of smaller Gaussians to accommodate finer-grained details. At each such iteration, the placement of new primitives is guided by a 3D error map, obtained by the back projection of 2D projections' residuals. This process acts as adaptive importance sampling in 3D space, directing model capacity to regions with high error. We evaluate our approach on sparse-view computed tomography reconstruction tasks, demonstrating improved performance over existing methods.

## 1 Introduction

Computed Tomography (CT) is a widely used imaging technology enabling non-destructive inspection of internal structures in various domains, *e.g.*, industrial quality control and medical diagnostics (Kak & Slaney, 2001; Herman, 2009). A key challenge in CT imaging is the trade-off between scan settings and image quality: more intensive scans provide detailed reconstructions but increase system usage and radiation exposure. To address this, recent research has focused on sparse-view CT reconstruction (Shen et al., 2022; Li et al., 2025; Zha et al., 2022; Xie et al., 2025; Cai et al., 2024b; Zha et al., 2024), which aims to perform accurate reconstruction from minimal projection data.

3D Gaussian Splatting (3DGS) (Kerbl et al., 2023) was first introduced as an explicit scene representation model for novel view synthesis under natural lighting conditions, and later adapted for CT reconstruction (Cai et al., 2024a; Zha et al., 2024; Wang et al., 2025). A fundamental aspect of the 3DGS model is how Gaussians are introduced and positioned. Current approaches typically rely on a dense, one-time initialization of primitives. This representation is subsequently refined through a densification strategy guided by accumulated 2D projection gradients. This process is local in its nature, since it splits or clones primitives only in the immediate vicinity of their parents. Additionally, the supervisory signal is indirect: a 2D projection gradient indicates that refinement is needed but does not uniquely specify the 3D location of the error. In sparse-view settings, this lack of direct 3D information leads to overfitting observed views, producing artifacts in unobserved regions.

In this work, we propose, instead, a 3D error-guided reconstruction approach within a hierarchical, layer-based framework (*cf.* Figure 1). In this formulation, a *layer*[1] refers to a new set of Gaussians added at a specific stage to address remaining volumetric errors. The process begins by initializing a coarse-grained layer of Gaussians representing the scene to capture the basic shape of the object. Subsequent layers of smaller and lower-density Gaussians are then iteratively introduced to resolve finer details and perform error correction. The core of our method is an error-driven strategy that exploits the known CT scanner geometry to aggregate 2D residuals from all views and reconstructs an explicit 3D error map. This map provides explicit guidance for both densification and sparsification: areas with positive error indicate *under-represented* regions requiring the inclusion of new Gaus-

---

[1]The term "layer" here is borrowed from the Computer Graphics field.

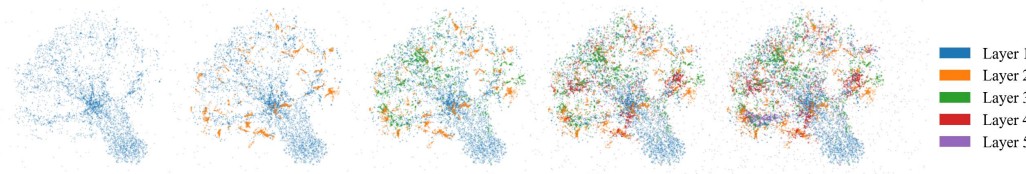

Figure 1: Overview of the layer-based approach. The volume is progressively reconstructed by adding and jointly optimizing new layers of Gaussians, each focusing on residual errors left by previous layers. The point clouds depict Gaussian centers, with colors indicating different layers.

sians, while negative error regions highlight *over-represented* areas that require merging existing Gaussians.

While our method is hierarchical, its core mechanism is fundamentally different from existing hierarchical approaches, *e.g.*, (Kerbl et al., 2024; Müller et al., 2022; Zha et al., 2022; Rückert et al., 2022), which typically organize primitives by dividing the volume into independent subunits (*e.g.*, octree nodes or hash grid cells). In those frameworks, primitives are locked to their assigned unit and optimized independently. This approach is often suboptimal in sparse-view CT because (i) many local cells have a too sparse ground truth signal, leading to inconsistent solutions at cell boundaries, and (ii) such structures typically rely on fixed subdivision rules that cannot easily recover if the initial partitioning is incorrect. Our method, instead, employs a *non-rigid hierarchy induced by residuals*, rather than a spatial one. In this sense, our approach is *holistic*: we optimize the representation globally rather than partitioning it into independent, localized sub-units. As a result, all layers coexist in the same 3D space and are jointly optimized to ensure global consistency.

This design is conceptually inspired by the coarse-to-fine approach (Tanimoto & Pavlidis, 1975; Burt & Adelson, 1987), often used in 3D reconstruction tasks (Gu et al., 2020; Yi et al., 2020; Wang et al., 2021; Barron et al., 2021; Yu et al., 2021; Kerbl et al., 2024), and the principle of iterative residual fitting. At its core is the idea that learning a corrective update to an existing representation is a more effective optimization strategy than learning the entire transformation from scratch. This principle empowers successful methods across different fields, including ensemble techniques such as Gradient Boosting (Friedman, 2001), and deep learning architectures such as Residual Networks (He et al., 2015). In our method, each layer of Gaussians is placed and optimized to correct the volumetric errors highlighted by the 3D error map generated in the previous layers. This step-by-step refinement reaches superior performance in the sparse view reconstruction.

The main contributions of this work can be summarized as follows: **(a)** we introduce a *hierarchical, layer-based* framework that approaches 3DGS reconstruction as an *iterative residual fitting* problem. Unlike standard local splitting strategies or rigid spatial partitioning, we employ a *holistic, non-rigid hierarchy* where new layers of primitives are globally initialized and jointly optimized to correct errors left by previous layers; **(b)** we propose a *3D error-driven guidance strategy* that exploits the known scanner geometry to reconstruct an explicit volumetric error map. This map provides direct structural guidance in 3D space, that enables distinct mechanisms for both *densification* (adding primitives in under-represented regions) and a novel concept of *sparsification* (fusing primitives in over-represented regions); **(c)** we demonstrate, with a series of experimental evaluations, that our method achieves state-of-the-art performance and geometric fidelity on the sparse-view CT reconstruction task. Furthermore, we validate our design choices through ablation studies.

## 2 RELATED WORK

### 2.1 TRADITIONAL RECONSTRUCTION

The foundational task in computed tomography, specifically for cone-beam CT, is the reconstruction of a volume from its 2D projections, a process framed as the inverse Radon transform (Kak & Slaney, 2001). Traditional methods to solve this problem fall into two main categories: analytical and iterative. The filtered backprojection algorithm FDK (Feldkamp et al., 1984) remains the standard analytical method for cone-beam CT reconstruction due to its computational efficiency and

simplicity. Iterative methods, including CGLS (Hestenes & Stiefel, 1952), SART, Andersen & Kak (1984), SART TV Biguri et al. (2016), ASD-POCS (Sidky & Pan, 2008), formulate reconstruction as an optimization problem, aiming to recover a volume that best explains the measured projections. However, traditional methods assume dense and high-quality projections. In sparse-view or noisy settings, they often produce artifacts and noise that reduce reconstruction accuracy.

## 2.2 CONTINUOUS REPRESENTATION

To address these limitations, recent methods have moved beyond traditional solvers by modeling the volume as a continuous field, using implicit neural representation (Sitzmann et al., 2020) and explicit representation with 3D Gaussian primitives (Kerbl et al., 2023). They follow two main strategies for 3D reconstruction: augmenting the input data via novel view synthesis or directly optimizing a continuous volumetric representation.

One line of work follows a two-stage strategy, where novel view synthesis (NVS) generates images from unseen viewpoints to augment the available projections for 3D reconstruction. Representative methods include neural radiance fields (Mildenhall et al., 2020; Zha et al., 2022; Cai et al., 2024b), modeling the scene as an implicit continuous function parameterized by neural network weights. Recently, 3DGS (Kerbl et al., 2023; Cai et al., 2024a) offers an explicit alternative, representing scenes with a collection of learnable Gaussians. These primitives are projected onto image planes through efficient splatting operations, enabling fast and high-fidelity rendering.

A more direct paradigm bypasses the intermediate NVS step, and instead learns a continuous representation of the volume that is optimized end-to-end from the sparse projections. Methods in this category include implicit neural representations (Zha et al., 2022; Xie et al., 2025; Shen et al., 2022) as well as explicit representations (Zha et al., 2024; Li et al., 2025), which have shown state-of-the-art performance. For example, 3DGR-CT (Li et al., 2025) renders projections by first voxelizing the Gaussian field into a 3D grid and then applying a differentiable CT projector. $R^2$-Gaussian (Zha et al., 2024) employs a custom radiative rasterizer, while using a voxelized grid solely to apply a total variation loss for regularization. Our work builds upon this direct, end-to-end paradigm, but introduces a hierarchical Gaussian representation guided by a 3D error map. This design enables iterative refinement of the volume via targeted updates, improving reconstruction accuracy in the sparse-view settings.

## 2.3 HIERARCHICAL REPRESENTATION

To improve computational efficiency and scalability, a common strategy is to structure the representation hierarchically. A prevalent approach involves spatial partitioning, where the scene is divided into independent sub-units. Prior work has employed multi-resolution feature grids to accelerate training (Müller et al., 2022; Zha et al., 2022), as well as adaptive tree structures such as the octrees (Martel et al., 2021; Rückert et al., 2022) or discrete structural primitives (Lu et al., 2024; Shen et al., 2025) to dynamically allocate model capacity. Other hierarchical designs include training pyramids of models for scale-aware rendering (Turki et al., 2023) and dividing the scene into spatial chunks for large-scale environments (Kerbl et al., 2024; Kulhanek et al., 2025).

In contrast to these approaches, our work adopts a holistic, non-rigid hierarchy. Instead of partitioning the space into independent chunks, we build the representation progressively through additive layers of Gaussian primitives. Importantly, each layer is guided by a global 3D error map to correct the volumetric residuals of the previous layers, allowing all primitives to be jointly optimized in a continuous space to ensure global consistency.

## 3 METHODOLOGY

A CT scan is the representation of an object (volume) through its *radiodensity field* $\sigma(\mathbf{x}) : \mathbb{R}^3 \to [0, 1]$ [2], associating with each coordinate $\mathbf{x}$ of the scanned volume an X-ray attenuation value representing the internal structure of the object. The primary goal of CT reconstruction is to recover the volume's radiodensity field leveraging a set of spatially localised 2D projections $\{(\mathbf{I}_v, \Gamma_v)\}_{v=1}^V$

---

[2]The radiodensity may have a different codomain, based on the application.

as the supervised signal. Each projection $\mathbf{I}_v : \mathbb{R}^{d_1 \times d_2} \to \mathbb{R}_{\geq 0}$ is a 2D X-ray image acquired from a specific viewpoint $v$. The geometry information $\Gamma_v$ specifies the acquisition parameters of the cone-beam scanner, including the source-to-detector distance, projection angle, detector pixel size, and more. The task consists, then, of devising an opportune representation of the volume whose projections taken from the same viewpoints coincide with the original projection. Our approach represents the volume leveraging a 3D Gaussian Splatting model, where a set of Gaussian primitives models the tomographic data reconstruction. Instead of training all the Gaussians at once, in our approach, we define a hierarchical representation where the Gaussians are divided into layers. New layers are progressively included and fit to the representation to mitigate the error "unmodeled" by previous ones. In the following, we describe our proposed methodology to perform reconstruction.

### 3.1 3D Gaussian representation for X-ray imaging

A popular approach to CT reconstruction represents the volume through a collection of Gaussian primitives $\{\mathcal{G}_i\}_{i=1}^N$. Each primitive $\mathcal{G}_i$ defines a localized distribution in space, which is geometrically described by a center position $\mu_i \in \mathbb{R}^3$ and a covariance matrix $\mathbf{\Sigma}_i \in \mathbb{R}_{\succeq 0}^{3 \times 3}$, controlling its shape and orientation. In essence,

$$\mathcal{G}_i(\mathbf{x}; \mu_i, \mathbf{\Sigma}_i) \propto \exp\left\{ -\frac{1}{2}(\mathbf{x} - \mu_i)^\top \mathbf{\Sigma}_i^{-1}(\mathbf{x} - \mu_i) \right\}. \tag{1}$$

The radiodensity field $\sigma(\mathbf{x})$ is then modeled as a linear combination of $N$ Gaussian primitives, each scaled by a corresponding central density parameter $\alpha_i \in [0, 1]$:

$$\sigma(\mathbf{x}) = \sum_{i=1}^N \alpha_i \, \mathcal{G}_i(\mathbf{x}; \mu_i, \mathbf{\Sigma}_i) \tag{2}$$

The model is therefore parameterised by $(\alpha_i, \mu_i, \mathbf{\Sigma}_i)_{i=1}^N$. To train these parameters, the 3D radiodensity field must be rendered into 2D projections that can be compared with the ground truth. This rendering process simulates the physics of X-ray imaging, which follows the Beer-Lambert law (Kak & Slaney, 2001). Specifically, the value of each pixel in a projection image corresponds to the line integral of the radiodensity field along the ray traced from the X-ray source to that pixel. For a single ray $\mathbf{r}(t)$, the projected value $\mathcal{I}(\mathbf{r})$ is given by:

$$\mathcal{I}(\mathbf{r}) = \int_{\mathbf{r}} \sigma(\mathbf{x}) \, \mathrm{d}t = \int_{\mathbf{r}} \sum_{i=1}^N \alpha_i \, \mathcal{G}_i(\mathbf{x}; \mu_i, \mathbf{\Sigma}_i) \, \mathrm{d}t = \sum_{i=1}^N \alpha_i \int_{\mathbf{r}} \mathcal{G}_i(\mathbf{x}; \mu_i, \mathbf{\Sigma}_i) \, \mathrm{d}t. \tag{3}$$

While the line integral defines the physical process, its direct computation is inefficient. Therefore, we employ a differentiable rasterization approach based on the principles of splatting (Zwicker et al., 2002; Kerbl et al., 2023). Specifically, we adopt the rasterization logic from the $\mathrm{R}^2$-Gaussian framework (Zha et al., 2024), designed for tomographic reconstruction. In this method, each 3D Gaussian is projected onto the 2D detector plane, and the final pixel values are computed by summing the contributions of these projected 2D Gaussians. This process yields a fully differentiable rendered projection, $\hat{\mathbf{I}}_v$, which can be compared to the ground truth image $\mathbf{I}_v$ for optimization.

### 3.2 Layer-based approach

Our reconstruction strategy adopts a layered architecture, whose pipeline is illustrated in detail in Figure 2. The process begins by initializing a base layer of Gaussians, $G^{(0)}$, by sampling from an initial volume created with a classical reconstruction method. At each subsequent layer $l \geq 1$, a new set of $N_\ell$ primitives is introduced and added to the cumulative model from all previous layers, $G^{(l-1)}$. The updated model is formed by the union $G^{(l)} = G^{(l-1)} \cup \{\mathcal{G}_j^{(l)}\}_{j=1}^{N_\ell}$, where the newly added Gaussians, $\{\mathcal{G}_j^{(l)}\}_{j=1}^{N_\ell}$, are strategically placed to correct the residual error left uncorrected by the previously optimized layers in $G^{(l-1)}$.

**3D error reconstruction** Let $\hat{\mathbf{I}}_v^{(l)}$ denote the rendered projection obtained with the first $l$ layers. We can then quantify the error of such representation as $\mathbf{e}_v^{(l)} = \mathbf{I}_v - \hat{\mathbf{I}}_v^{(l)}$. Notably, instead of directly

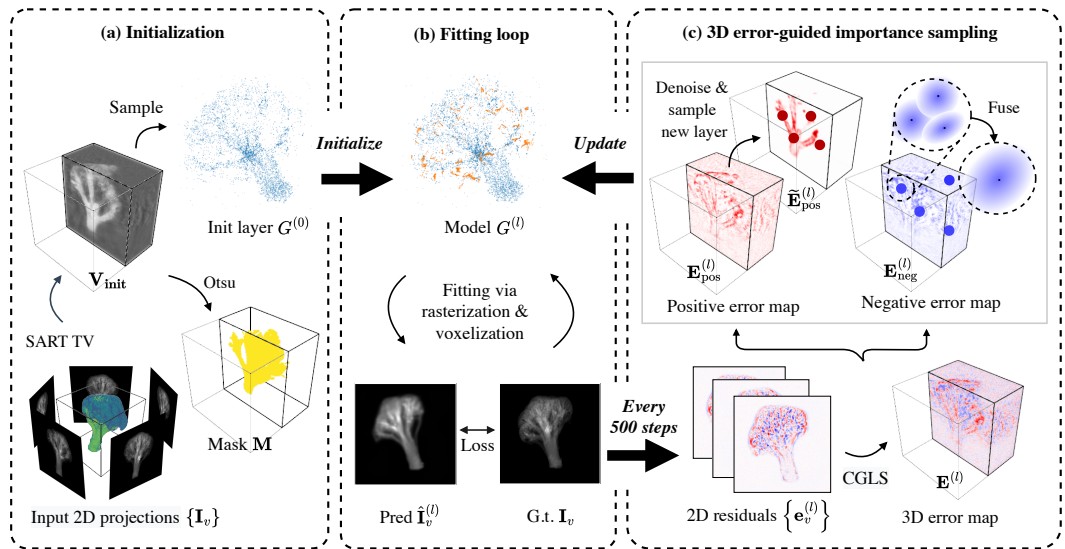

Figure 2: Overview of the layer-based reconstruction pipeline. (a) A classical tomographic reconstruction generates an initial volume to guide the sampling of the first Gaussian layer and to compute an object mask. (b) The iterative loop renders the current set of Gaussians via rasterization and voxelization to produce predicted projection images, which are compared with the ground truth to compute residual maps. (c) These residuals are reconstructed into a 3D volumetric error map, which guides an importance sampling strategy: positive-error regions are sampled for densification (adding a new layer of Gaussians), while negative-error regions are sampled for sparsification (fusing existing Gaussians). The properties of the new Gaussians are adaptively initialized based on the local error magnitude and the current model state.

optimizing the Gaussian positions in the 2D projection space, we use the residual maps $\mathbf{e}_v^{(l)}$ from different views to solve an inverse problem, yielding a 3D volumetric error map $\mathbf{E}^{(l)}(\mathbf{x})$ with the same size as the original volume. The 3D error reconstruction is achieved by solving the linear least-squares problem using the Conjugate Gradient for Least Squares (CGLS) (Hestenes & Stiefel, 1952; Biguri et al., 2016).

**Sampling procedure**  To guide the model's refinement, we first decompose the 3D error map into its positive and negative components. We then independently sample locations from each map using the Gumbel-Max trick (Gumbel, 1958), which correspondingly guide densification and sparsification (*cf.* Figure 2). The sampling is error-guided and explained in detail in Appendix B. Based on the sign of the error at the sampled locations, two complementary updates can be performed. Positive error regions indicate *density under-representation*, where the model lacks sufficient representation. We densify these regions by placing new Gaussian primitives, thereby enriching the scene with finer detail and improving reconstruction fidelity (*cf.* Figure 1). Negative error regions reveal *density over-representation*, where too many Gaussians contribute additional value. In these areas, we reduce density through local fusion, merging nearby Gaussians into a single, less dense primitive. This acts as a form of regularization by reallocating capacity and reducing geometric redundancy. In the following, we describe both operations in detail.

**Layered densification**  In the case of a location associated with positive error, we place a new Gaussian $\mathcal{G}_i^{(l)}$, whose density $\alpha_i^{(l)}$ is initialized in proportion to the local error $e_i^{(l)} = \mathbf{E}^{(l)}(\mathbf{x}_i)$ at sampled point $\mathbf{x}_i$. As the model grows, we normalize this initial density by the model's current capacity, approximated by the number of primitives $N^{(l-1)}$, to ensure that new primitives provide gentle corrections to the residual error rather than destabilizing the structure established by previous layers. We apply the following scaling, tuned by the hyperparameter $C_\alpha$, to ensure a stable

contribution in the final rendering:

$$\alpha_i^{(l)} = C_\alpha \frac{e_i^{(l)}}{\sqrt[3]{N^{(l-1)}}}. \tag{4}$$

The initial size of new Gaussians is set to a fraction of the average scale of all existing primitives present in the model. This enforces a coarse-to-fine refinement strategy: as the model's average scale naturally decreases with each layer, new primitives are born progressively smaller, ensuring they are dedicated to capturing finer details rather than re-learning the established broad structure.

**Layered sparsification** In case of a location associated with negative error, we sample a set of points from these regions to serve as fusion centers. At each center, we fuse all Gaussians within a small neighborhood $\mathcal{N}_\varepsilon(i)$ into a single primitive. The density $\alpha_i^{(l)}$ of the fused Gaussian is calculated by summing the densities of the neighbors and then reducing the total by the local error (in absolute value) scaled by the model's capacity, similar to the initialization in Equation 4:

$$\alpha_i^{(l)} = \sum_{j \in \mathcal{N}_\varepsilon(i)} \alpha_j^{(l-1)} - C_\alpha \frac{\left| e_i^{(l)} \right|}{\sqrt[3]{N^{(l-1)}}}. \tag{5}$$

Other properties are aggregated based on neighbor density: position and scale are computed via a weighted average, rotation is inherited from the most opaque neighbor. This ensures the fused primitive represents the dominant local structure while avoiding the complexities of rotational averaging.

### 3.3 TRAINING

Guided by this 3D error map, we add a new layer of Gaussian primitives, $\mathcal{G}^{(l+1)}$, strategically placed within high-error regions. After that, the updated model is jointly optimized to minimize the reconstruction error across all projections using a differentiable projection-domain loss function $\mathcal{L}_{\text{total}}$. Following Zha et al. (2024), we compute the optimization loss and update our model on a per-view basis, progressively selecting them in random order:

The total loss $\mathcal{L}_{\text{total}}$ comprises a photometric $\mathcal{L}_1$ and a structural fidelity $\mathcal{L}_{\text{SSIM}}$ term. Similar to Zha et al. (2024), we incorporate a 3D total variation $\mathcal{L}_{\text{TV}}$ regularization term applied to randomly sampled volumetric patches. This term imposes a smoothness prior by penalizing high-frequency variations in the radiodensity field:

$$\mathcal{L}_{\text{total}}(\mathbf{I}_v, \hat{\mathbf{I}}_v^{(l)}) = \mathcal{L}_1(\mathbf{I}_v, \hat{\mathbf{I}}_v^{(l)}) + \lambda_{\text{SSIM}} \mathcal{L}_{\text{SSIM}}(\mathbf{I}_v, \hat{\mathbf{I}}_v^{(l)}) + \lambda_{\text{TV}} \mathcal{L}_{\text{TV}}(\boldsymbol{X}_p; G^{(l)}), \tag{6}$$

where $\lambda_{\text{SSIM}}$ and $\lambda_{\text{TV}}$ are two hyperparameters, and $\boldsymbol{X}_p$ is a 3D patch randomly sampled from the current model $G^{(l)}$ used to compute the total variation loss.

## 4 EXPERIMENTS

In this section, we describe the experimental setup used to assess the performance of our method and discuss the results we have obtained. Additionally, we conduct several ablation studies to gain additional insight into the motivation behind the reported performance.

### 4.1 EXPERIMENTAL SETTINGS

**Dataset** Following (Cai et al., 2024a; Zha et al., 2024), we conduct experiments on both synthetic and real-world datasets representing diverse object types. The synthetic dataset includes fifteen 3D volumes categorized into three classes: medical, food, and everyday objects. We use the TIGRE (Biguri et al., 2016) toolbox to generate cone-beam X-ray projections, simulating realistic imaging conditions by incorporating Compton scatter and electronic noise. The real-world dataset includes scans of walnut, seashell, and pine. Similarly, we treat the fully reconstructed high-resolution volumes of these objects as ground truth and simulate a sparse-view acquisition using the same projection pipeline as the synthetic set.

Table 1: Quantitative comparison on the 3D reconstruction task across 5, 10, 15, and 25 views settings. The reported metrics are computed over the full volumes and averaged across all scans. We apply colors to the first , second , and third ranked numbers.

| Methods | 5 views | | | 10 views | | | 15 views | | | 25 views | | |
|---|---|---|---|---|---|---|---|---|---|---|---|---|
| | PSNR↑ | SSIM↑ | Time↓ | PSNR↑ | SSIM↑ | Time↓ | PSNR↑ | SSIM↑ | Time↓ | PSNR↑ | SSIM↑ | Time↓ |
| Real dataset | | | | | | | | | | | | |
| FDK | 14.71 | 0.066 | – | 17.77 | 0.106 | – | 19.34 | 0.138 | – | 23.30 | 0.335 | – |
| CGLS | 24.57 | 0.546 | 0.7s | 26.21 | 0.585 | 0.7s | 27.18 | 0.611 | 0.8s | 28.26 | 0.673 | 0.9s |
| SART | 25.84 | 0.648 | 6.1s | 28.21 | 0.696 | 11.0s | 29.61 | 0.722 | 16.0s | 31.52 | 0.790 | 26.0s |
| SART TV | 26.65 | 0.720 | 33.0s | 29.68 | 0.795 | 54.7s | 31.24 | 0.829 | 1.3m | 32.89 | 0.836 | 2.1m |
| ASD-POCS | 26.65 | 0.711 | 43.6s | 29.84 | 0.788 | 51.6s | 31.60 | 0.827 | 52.2s | 32.38 | 0.826 | 2.0m |
| NAF | 27.94 | 0.802 | 1.3m | 32.47 | 0.859 | 2.5m | 33.90 | 0.876 | 3.7m | 32.76 | 0.783 | 7.2m |
| X-Gaussian | 20.72 | 0.639 | 3.1m | 20.73 | 0.637 | 3.0m | 20.73 | 0.638 | 2.9m | 20.72 | 0.636 | 5.4m |
| R2-Gaussian | 27.24 | 0.715 | 4.3m | 31.90 | 0.812 | 4.7m | 34.40 | 0.854 | 4.9m | 35.52 | 0.843 | 7.7m |
| **Ours** | 28.75 | 0.828 | 5.6m | 33.59 | 0.891 | 6.3m | 35.47 | 0.908 | 6.6m | 36.46 | 0.850 | 8.4m |
| Synthetic dataset | | | | | | | | | | | | |
| FDK | 12.66 | 0.045 | – | 15.26 | 0.068 | – | 16.81 | 0.090 | – | 22.99 | 0.317 | – |
| CGLS | 22.79 | 0.482 | 0.7s | 24.64 | 0.512 | 0.7s | 25.61 | 0.535 | 0.8s | 27.99 | 0.664 | 0.9s |
| SART | 24.10 | 0.638 | 5.7s | 26.31 | 0.669 | 10.7s | 27.58 | 0.683 | 15.5s | 31.14 | 0.825 | 25.4s |
| SART TV | 24.88 | 0.709 | 31.6s | 27.70 | 0.766 | 55.6s | 29.20 | 0.795 | 1.3m | 31.48 | 0.864 | 2.3m |
| ASD-POCS | 24.98 | 0.725 | 42.4s | 27.91 | 0.779 | 45.9s | 29.52 | 0.806 | 47.7s | 33.92 | 0.907 | 1.6m |
| NAF | 25.11 | 0.724 | 1.2m | 28.29 | 0.781 | 2.4m | 29.82 | 0.804 | 3.6m | 33.48 | 0.893 | 6.1m |
| X-Gaussian | 17.45 | 0.620 | 3.5m | 17.46 | 0.620 | 3.3m | 17.46 | 0.620 | 3.1m | 17.46 | 0.620 | 3.7m |
| R2-Gaussian | 23.48 | 0.670 | 9.8m | 27.00 | 0.759 | 8.4m | 29.60 | 0.813 | 7.6m | 35.39 | 0.926 | 5.7m |
| **Ours** | 25.67 | 0.788 | 5.8m | 29.65 | 0.858 | 6.4m | 31.62 | 0.886 | 6.9m | 34.43 | 0.922 | 7.5m |

**Implementation details** We implement our method in PyTorch with CUDA acceleration [3]. All experiments are conducted on an NVIDIA H200 GPU. As initialization step, we reconstruct a coarse volume using the SART-TV algorithm (Biguri et al., 2016). A binary object soft mask is then calculated using Otsu thresholding (Otsu, 1979). Next, we construct the Gaussian field using a 20-layer hierarchical strategy, where a new layer of 2500 Gaussians is introduced every 500 iterations, guided by the reconstructed 3D error map. The positive error map is first denoised using the object mask and a 3D Gaussian blur with a standard deviation of $\sigma = 2$ to yield an importance map. From this map, we sample locations for new Gaussians using the Gumbel-max trick with temperature $\tau = 5 \times 10^{-3}$. To progressively capture finer details, these new Gaussians are initialized with a reduced scale (half the current average) and a density scaled by $C_\alpha = 0.5$ (Equations (4–5)). Concurrently, the negative error map guides our layered sparsification. We sample 30K fusion centers and aggregate nearby Gaussians within a radius of $\epsilon = 0.05$ into a single, less dense primitive. This layer-building phase continues until all layers are placed, with all existing Gaussians being jointly optimized. Afterwards, vanilla density control (Kerbl et al., 2023) is enabled for a final fine-tuning stage. Across all experiments, the total optimization runs for 30K iterations. To quantitatively assess reconstruction quality, we use the 3D PSNR and SSIM metrics, following the implementation provided in R$^2$-Gaussian (Zha et al., 2024). More details can be found in Appendices A and B.

**Baselines** We benchmark our method against traditional reconstruction techniques, as well as recent implicit and explicit representation techniques. This comparison is limited to self-supervised approaches, as our work focuses on single-scene reconstruction, where supervised methods requiring external datasets are not applicable. As classical baselines, we use the analytical FDK algorithm (Feldkamp et al., 1984) and several iterative methods. While many variations of iterative reconstruction algorithms exist, we focus on a few representative ones: CGLS (Hestenes & Stiefel, 1952), SART (Andersen & Kak, 1984), SART-TV (Biguri et al., 2016), and ASD-POCS (Sidky & Pan, 2008). We tune the number of iterations for each classical algorithm to achieve optimal results

---
[3]Our code is available at the anonymous GitHub repository

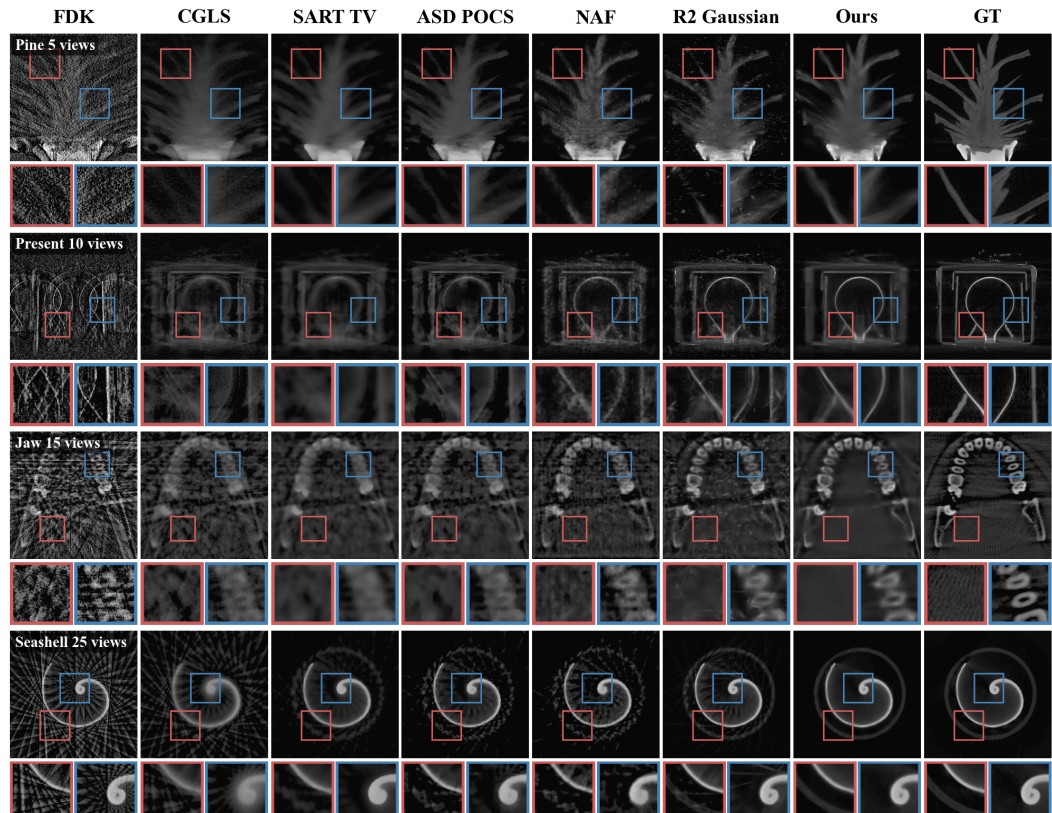

Figure 3: Qualitative evaluation of reconstruction algorithms under varying degrees of data sparsity. Our coarse-to-fine approach uses a 3D volumetric error map to strategically allocate model capacity, focusing refinement only on regions with high structural error. This targeted strategy avoids overfitting to sparse views, resulting in a cleaner and more geometrically accurate reconstruction. Zoomed-in patches are provided for a clearer inspection of reconstruction quality.

across all datasets. Alongside traditional approaches, we evaluate recent implicit and explicit reconstruction methods. For implicit reconstruction, we use NAF (Zha et al., 2022), which models voxel-wise attenuation coefficients as continuous neural fields. For explicit reconstruction, we consider X-Gaussians (Cai et al., 2024a) and $R^2$-Gaussian (Zha et al., 2024) models. Since X-Gaussians is designed for novel view synthesis, we first generate intermediate projections and then apply the classical reconstruction algorithm ASD-POCS to obtain the volume. R-Gaussian, in contrast, reconstructs the volume directly using voxelization.

## 4.2 BASELINE COMPARISON

The performance of our layer-based method is evaluated both quantitatively and qualitatively. Table 1 reports the volumetric 3D PSNR and SSIM metrics for various sparse-view configurations, where our method consistently demonstrates improved performance. More detailed results are provided in Appendix C. These numerical gains are visually supported in Figure 3. The qualitative comparison shows that our layer-based Gaussian model achieves higher fidelity, producing cleaner transitions between neighboring views and preserving finer details.

**Analysis of reconstruction accuracy**   Our method demonstrates its main strengths in sparse-view settings (*e.g.*, 5-15 views), while comparable in more densely-sampled scenarios. This advantage stems from our progressive, coarse-to-fine strategy, which acts as a regularizer against overfitting. The initial layers establish a robust, low-frequency structure using large Gaussians, capturing the object's general form. Subsequent refinements are guided by the 3D volumetric error map, which

strategically places smaller, lower-density Gaussians to correct true volumetric inaccuracies. This ensures better geometric fidelity and generalization to in-between views. Conversely, in more data-rich settings (*e.g.*, 25 views), the large number of projections provides more reliable signals from all directions. This is often sufficient to effectively guide a standard optimization of all Gaussians at once. In these cases, the implicit regularization from our layered approach is less critical.

**Analysis of runtime** Our method's overall timing is comparable with state-of-the-art approaches. Our process introduces extra computations from the initial SART-TV reconstruction and the periodic layered densification and sparsification. However, these costs are effectively balanced by the efficiency gains from our layered fitting strategy. Instead of optimizing all $N$ Gaussians from the start, our method incrementally builds the scene in $L$ stages. For a significant portion of the training, we operate on a much smaller subset of primitives, each time including $N/L$ Gaussians. Furthermore, our layered sparsification step prunes redundant Gaussians. The combined effect produces a more compact representation and comparable fitting time, as empirically validated in Table 3.

### 4.3 Ablation study

**Layered densification** We investigate how reconstruction quality is affected by the choice in the placement of new Gaussian layers. Specifically, we focus on the hierarchical depth, defined by the number of layers, and (2) the spatial density, defined by the initial number of Gaussians per layer. We set a total of 50K primitives to be introduced during the layer-building phase. In a multi-layer setup with $L$ layers, primitives are added incrementally in batches of $50\text{K}/L$ per layer. This is compared against a single-layer baseline where all 50K primitives are present from the start. Table 3 presents results for different architectures of layered Gaussian model. We report 3D metrics for reconstruction quality, the total number of Gaussians after 30K optimization steps, and training time. Results show that generally multi-layered approaches outperform the single-layer baseline. Moreover, multi-layer design achieves this performance with less number of primitives and, therefore, less training time, with the 20 layer configuration achieving the best results across all views. Figure 4 provides visual comparison of multi-layer and one-layer approaches.

**Layered sparsification** We analyze the key hyperparameters of our sparsification mechanism, the number of sampled fusion centers and the fusion radius, which together control the degree of structural regularization. This volumetric, error-guided fusion is distinct from standard density-based pruning. Its goal is to manage model complexity by correcting for over-represented regions identified in the 3D error map. An ablation study was performed to identify the optimal configuration that maximizes reconstruction accuracy, as detailed in Table 4, revealing the need to balance between overfitting from insufficient fusion and over-smoothing from an overly aggressive approach.

**Masking** We analyze the impact of applying the soft Otsu mask during our error-guided densification. As shown in Table 5, the mask provides a spatial prior that yields improvements in both reconstruction quality and model compactness. Without this prior, the model tends to place Gaussians in empty space to minimize 2D projection errors, a form of overfitting that degrades the 3D structure and inflates model size. By constraining densification to the object's volume, the mask ensures model capacity is used to refine true geometric details. This results in superior 3D metrics and a more regularized model with substantially fewer Gaussians.

**Layer selection** We explore training strategies to mimic a boosting-like approach: train only the newest layer, train the last few layers, and probabilistically select layers, either as a contiguous chain or independently. Although these methods can reduce computation, optimizing all layers consistently produces the best results as shown in Table 6.

## 5 Discussion

Our framework demonstrates a robust approach to sparse-view reconstruction, yet it leaves room for future exploration. First, the accuracy of the guiding 3D error map is tied to the quality of the back-projection solver. In highly sparse scenarios, this map can become noisy, especially after many layers, potentially leading to the placement of new primitives that capture noise artifacts instead of

true structural errors. We mitigate this by denoising the error map with soft object mask and 3D Gaussian blur, but more advanced techniques could be explored in the future. Second, a promising direction is to move beyond a fixed number of primitives per layer towards an adaptive strategy where the number and properties of new Gaussians are determined theoretically by the local error distribution. Third, beyond CT data, this concept may generalize to other tomographic modalities, highlighting the broader relevance of explicit error-guided reconstruction. Finally, the core principle of our method is fundamentally representation-agnostic. For example, the 3D error map can guide importance sampling of training coordinates for an implicit neural representation, directing the network's capacity toward high-error regions and mirroring the coarse-to-fine refinement strategy.

## 6 CONCLUSION

In this work, we introduced a hierarchical, layer-based coarse-to-fine framework for sparse-view CT reconstruction leveraging a 3D error map to guide the iterative refinement of a 3D Gaussian representation. Our densification and sparsification strategy allocates model capacity more effectively by directly addressing volumetric inaccuracies. This mitigates a key problem in baseline methods, often overfitting to the training projections when initialized with a dense set of primitives. As shown in our experiments, our approach yields reconstructions with superior geometric fidelity, particularly in highly sparse settings. Therefore, the principle of explicit 3D error correction offers a promising path towards more robust and reliable CT reconstruction in data-limited scenarios.

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

## A    MODEL INITIALIZATION

### A.1    INITIAL RECONSTRUCTION

For the initial approximation of the object volume, we employ the Simultaneous Algebraic Reconstruction Technique with Total Variation regularization (SART-TV) using the TIGRE toolkit (Biguri et al., 2016). Other approaches include uniform distribution initialization (Cai et al., 2024a), reconstruction with FDK (Zha et al., 2024), and mixed methods (Wang et al., 2025). We select SART-TV for its superior ability to produce a high-fidelity volume with well-defined edges and reduced artifacts for sampling the first layer and calculating the object mask.

### A.2    FIRST LAYER INITIALIZATION

To maintain a consistent sampling methodology throughout our layer-based framework, the initialization of the first layer of Gaussians, $G^{(0)}$, also employs an importance sampling strategy based on the Gumbel-Max trick. Unlike subsequent layers, which are guided by a 3D error map, the first layer is guided by the initial volumetric reconstruction produced by the SART-TV solver. This initial volume serves as a coarse density map, where voxel intensities represent the probability of belonging to the object. Although our method is more robust to initialization due to its iterative, error-correcting design, this starting estimate is still important for generating the object mask via Otsu thresholding used in subsequent denoising steps.

### A.3    OTSU SOFT MASKING

The mask is computed once at the beginning of the pipeline. To separate the object from the background, we apply Otsu's thresholding (Otsu, 1979) to determine the optimal binary threshold $t^*$:

$$t^* = \arg \max_t \mathrm{Var}_b(t), \tag{7}$$

where $\mathrm{Var}_b(t)$ denotes the between-class variance for a given threshold $t$.

In a sparse-view setting, the initial volume is prone to artifacts, which makes a traditional binary mask created by hard thresholding unreliable. Such a mask would incorrectly classify uncertain regions, potentially removing parts of the object or leaving background noise unfiltered. To mitigate these issues, we replace the binary mask with a probabilistic *soft* mask. First, for each voxel $\mathbf{x}$, we calculate a normalized and scaled distance $d(\mathbf{x})$ from the Otsu threshold $t^*$:

$$d(\mathbf{x}) = \beta \frac{\mathbf{V}_{\mathrm{init}}(\mathbf{x}) - t^*}{\sigma_V}, \tag{8}$$

where $\beta > 0$ is a steepness parameter and $\sigma_V$ is the standard deviation of the volume's intensities. This step quantifies how far each voxel is from the decision boundary.

Second, we apply the logistic sigmoid function to map this distance into a probabilistic value:

$$\mathbf{M}(\mathbf{x}) = \frac{1}{1 + \exp(-d(\mathbf{x}))}. \tag{9}$$

This formulation produces a smooth mask with values in $[0, 1]$, representing the probability of a voxel belonging to the object. This method preserves uncertain boundary regions and provides a more reliable guide for subsequent processing steps.

## B    SAMPLING PROCEDURE

To guide the placement of new Gaussians (densification) and the fusion of existing ones (sparsification), we require a robust method for sampling locations from the 3D error map, $\mathbf{E}^{(l)}$. Our procedure begins by decomposing this map into its positive and negative components, $\mathbf{E}_{\mathrm{pos}}^{(l)}$ and $\mathbf{E}_{\mathrm{neg}}^{(l)}$, which are sampled independently. The positive map, which guides densification, undergoes a denoising step prior to sampling to ensure new primitives are placed in regions of coherent error rather than noise. The negative map is sampled directly to identify candidates for fusion. For both maps, we employ an error-guided importance sampling strategy.

### B.1 Error-guided Gumbel sampling

To sample positions from either the positive or negative error maps, we use the Gumbel-Max trick (Gumbel, 1958). This technique allows for efficient importance sampling from a discrete distribution defined by unnormalized scores. This is particularly beneficial in large 3D volumes where computing a partition function would be computationally expensive.

Let $e_i^{(l)} = \mathbf{E}^{(l)}(\mathbf{x}_i)$ denote the absolute error value at position $\mathbf{x}_i$, and let $\tau$ be a temperature parameter controlling the stochasticity of the process. We generate Gumbel noise $g_i$ from the standard Gumbel distribution $g_i = -\log(-\log(u_i))$, where $u_i \sim \text{Uniform}(0, 1)$. The final score for position $\mathbf{x}_i$ is computed as $s_i^{(l)} = |e_i^{(l)}|/\tau + g_i$. We then select the top-$k$ highest-scoring indices $i_1, \ldots, i_k$ to form the set of sampled locations for layer $l$.

### B.2 Denoising of the error map

The error volume reconstructed from sparse-view data, $\mathbf{E}^{(l)}$, often contains artifacts such as streaks and noise. To ensure that new Gaussians are placed in regions of meaningful error rather than artifacts, we apply a two-stage denoising process to the positive component, $\mathbf{E}_{\text{pos}}^{(l)}$, before sampling. First, we apply a probabilistic soft mask, $\mathbf{M}(\mathbf{x})$, via a Hadamard product: $\widetilde{\mathbf{E}}_{\text{pos}}^{(l)} = \mathbf{M} \odot \mathbf{E}_{\text{pos}}^{(l)}$. The generation of this mask is detailed in Appendix A.3. Second, the masked volume is smoothed with a Gaussian blur. This step suppresses high-frequency noise and enhances spatial coherence, yielding a robust importance map that guides the subsequent Gumbel sampling for densification.

## C Per-scene comparison

We compare our layer-based approach with the $\mathrm{R}^2$-Gaussian model in Table 2. To ensure a fair comparison on the sparse-view datasets, we address overfitting issue in standard $\mathrm{R}^2$-Gaussian model (Zha et al., 2024). $\mathrm{R}^2$-Gaussian model trained with its original parameters tends to overfit to sparse views, resulting in severe needle-like artifacts, especially in settings with very few input images. To mitigate this effect and establish a stronger baseline, we adjusted parameters in favor to the sparse-view setting. Specifically, we (1) increased Total Variation regularization with a weight of $\lambda_{\text{TV}} = 0.25$ to encourage smoother geometry; (2) increasing the minimum allowed Gaussian scale to 0.005 to prevent overly thin structures; and (3) increased the densification gradient threshold to 0.001 to reduce excessive splitting and cloning. Collectively, these changes regularize the model and improve robustness under sparse-view conditions. However, they introduce a trade-off: a less-regularized model achieves higher fidelity in higher-view settings (e.g., 25 views), while the more-regularized model tends to oversmooth results and lowers the metrics. Despite this, our layer-based strategy achieves better results.

## D Ablations

### D.1 Layered densification

In Table 3, we present an ablation on the number of layers across different sparse-view settings. We report 3D PSNR, 3D SSIM, the number of Gaussians, and training time. Multi-layer architectures generally outperform the single-layer baseline while using fewer primitives and less training time. A configuration with 20 layers results in the best trade-off across all view settings.

### D.2 Layered sparsification

In Table 4, we present an ablation on sparsification hyperparameters, fusion radius, and the number of sampled fusion centers, comparing performance on real and synthetic datasets across different sparse-view settings. We report 3D PSNR, 3D SSIM, the number of Gaussians, and training time. The best parameters are highlighted. In Figure 4, we additionally include a visual comparison between the 1-layer and 20-layer models.

Table 2: 3D PSNR comparison between R2-Gaussian and our method across different numbers of sparse views on synthetic and real datasets. Gray-colored numbers indicate R2-Gaussian metrics obtained with a set of parameters optimized for the sparse-view setting.

| Scene | 5 views | | 10 views | | 15 views | | 25 views | |
|---|---|---|---|---|---|---|---|---|
| | R2-Gaussian | Ours | R2-Gaussian | Ours | R2-Gaussian | Ours | R2-Gaussian | Ours |
| Real dataset | | | | | | | | |
| Pine | 29.93 / 31.49 | 32.04 | 33.94 / 35.26 | 35.88 | 36.52 / 37.03 | 37.68 | 38.10 / 37.70 | 37.84 |
| Seashell | 29.00 / 29.04 | 30.97 | 34.99 / 34.73 | 37.43 | 37.81 / 36.57 | 39.24 | 39.53 / 38.77 | 41.52 |
| Walnut | 22.79 / 23.15 | 23.24 | 26.76 / 26.97 | 27.47 | 28.87 / 28.74 | 29.47 | 28.94 / 29.91 | 30.01 |
| **Average** | 27.24 / 27.89 | **28.75** | 31.90 / 32.32 | **33.59** | 34.40 / 34.11 | **35.47** | 35.52 / 35.46 | **36.46** |
| Synthetics dataset | | | | | | | | |
| Chest | 19.39 / 22.43 | 22.88 | 22.58 / 26.38 | 26.44 | 26.53 / 28.07 | 28.44 | 32.18 / 30.50 | 31.48 |
| Foot | 22.63 / 24.54 | 24.58 | 25.91 / 27.15 | 27.60 | 27.78 / 28.86 | 29.18 | 30.38 / 30.26 | 30.53 |
| Head | 24.04 / 26.49 | 26.97 | 28.79 / 30.77 | 31.14 | 30.77 / 32.10 | 32.55 | 36.86 / 35.68 | 36.34 |
| Jaw | 24.45 / 24.57 | 25.06 | 27.20 / 27.48 | 28.83 | 29.31 / 29.49 | 30.94 | 33.35 / 32.60 | 33.47 |
| Pancreas | 22.01 / 25.16 | 25.40 | 25.61 / 27.15 | 27.49 | 28.43 / 29.07 | 29.38 | 33.08 / 31.01 | 32.39 |
| Beetle | 32.52 / 32.88 | 32.94 | 34.98 / 34.22 | 34.65 | 37.39 / 35.45 | 36.41 | 40.09 / 36.12 | 37.36 |
| Bonsai | 21.68 / 24.92 | 25.10 | 23.78 / 28.33 | 28.50 | 26.18 / 29.88 | 30.00 | 33.06 / 32.03 | 32.56 |
| Broccoli | 18.37 / 19.80 | 19.95 | 20.95 / 22.48 | 22.54 | 23.13 / 24.96 | 25.01 | 29.25 / 28.16 | 28.62 |
| Kingsnake | 31.80 / 33.58 | 34.02 | 35.33 / 36.22 | 36.73 | 36.19 / 36.47 | 36.84 | 39.03 / 37.22 | 37.89 |
| Pepper | 16.25 / 17.97 | 18.42 | 20.16 / 26.24 | 26.43 | 24.21 / 28.76 | 29.16 | 35.08 / 32.52 | 34.21 |
| Backpack | 26.60 / 27.75 | 28.15 | 29.13 / 29.30 | 29.96 | 31.07 / 30.19 | 31.07 | 34.97 / 31.17 | 32.70 |
| Engine | 17.65 / 20.27 | 20.42 | 22.27 / 24.71 | 25.27 | 27.10 / 29.27 | 30.24 | 35.23 / 33.45 | 35.08 |
| Mount | 19.99 / 24.73 | 24.37 | 21.58 / 30.50 | 30.46 | 25.00 / 32.47 | 33.32 | 37.39 / 35.58 | 36.92 |
| Present | 26.10 / 26.27 | 26.88 | 28.73 / 28.24 | 29.35 | 30.26 / 29.14 | 30.37 | 35.04 / 31.18 | 33.42 |
| Teapot | 28.77 / 29.33 | 29.98 | 37.91 / 38.58 | 39.43 | 40.70 / 41.10 | 41.31 | 45.81 / 43.47 | 43.51 |
| **Average** | 23.48 / 25.38 | **25.67** | 27.00 / 29.18 | **29.65** | 29.60 / 31.02 | **31.62** | 35.39 / 33.40 | 34.43 |

Table 3: Ablation on the number of layers ($L$) across different sparse-view settings. Multi-layer architectures generally outperform the single-layer baseline while using fewer primitives ($N$) and less training time. $L = 20$ results in the best trade-off across all view settings.

| | $L$ | Real Dataset | | | | Synthetic Dataset | | | |
|---|---|---|---|---|---|---|---|---|---|
| | | PSNR↑ | SSIM↑ | $N$↓ | Time↓ | PSNR↑ | SSIM↑ | $N$↓ | Time↓ |
| 5 views | 1 | 27.68 | 0.773 | 50K | 8.1m | 25.46 | 0.765 | 66K | 8.7m |
| | 5 | 28.07 | 0.794 | 23K | 6.1m | 25.58 | 0.778 | 54K | 7.7m |
| | 10 | 28.18 | 0.801 | 19K | 5.5m | 25.61 | 0.781 | 39K | 7.0m |
| | 20 | 28.34 | 0.806 | 17K | 5.1m | 25.68 | 0.786 | 31K | 6.3m |
| | 30 | 28.37 | 0.809 | 15K | 4.9m | 25.73 | 0.788 | 28K | 5.8m |
| 10 views | 1 | 31.79 | 0.849 | 50K | 8.5m | 28.97 | 0.839 | 56K | 9.2m |
| | 5 | 32.33 | 0.866 | 32K | 7.0m | 29.16 | 0.849 | 45K | 8.1m |
| | 10 | 32.47 | 0.872 | 26K | 6.4m | 29.20 | 0.850 | 40K | 7.5m |
| | 20 | 32.68 | 0.880 | 22K | 6.1m | 29.30 | 0.852 | 37K | 7.0m |
| | 30 | 32.40 | 0.875 | 19K | 6.0m | 29.24 | 0.848 | 33K | 6.7m |
| 15 views | 1 | 33.62 | 0.880 | 50K | 8.6m | 30.76 | 0.870 | 54K | 9.3m |
| | 5 | 34.10 | 0.894 | 37K | 7.5m | 30.91 | 0.877 | 47K | 8.3m |
| | 10 | 34.21 | 0.898 | 31K | 7.1m | 30.93 | 0.878 | 43K | 7.8m |
| | 20 | 34.29 | 0.900 | 27K | 6.6m | 30.99 | 0.878 | 41K | 7.4m |
| | 30 | 33.84 | 0.891 | 24K | 6.5m | 30.87 | 0.874 | 37K | 7.0m |
| 25 views | 1 | 36.23 | 0.862 | 55K | 9.6m | 33.34 | 0.909 | 54K | 7.3m |
| | 5 | 36.46 | 0.856 | 52K | 8.1m | 33.42 | 0.912 | 49K | 6.5m |
| | 10 | 36.49 | 0.855 | 48K | 7.5m | 33.45 | 0.913 | 45K | 6.1m |
| | 20 | 36.38 | 0.853 | 47K | 6.8m | 33.46 | 0.913 | 42K | 5.6m |
| | 30 | 33.95 | 0.812 | 32K | 6.0m | 32.95 | 0.904 | 35K | 5.3m |

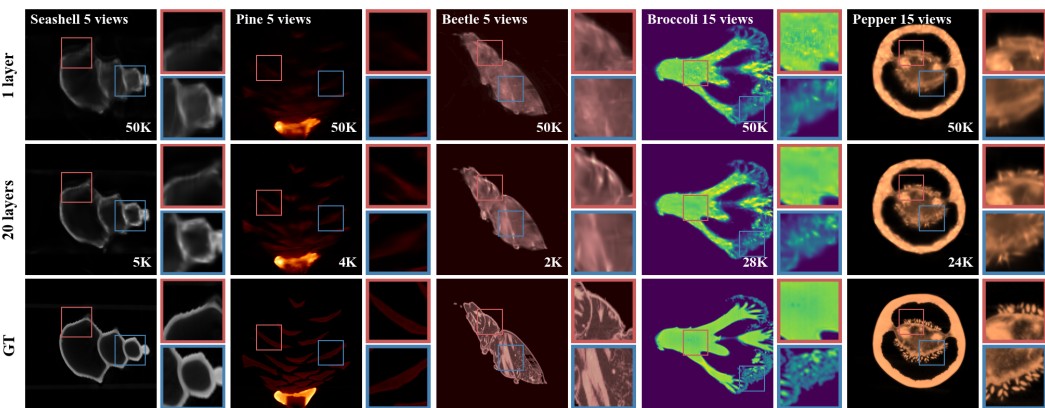

Figure 4: Comparison between 1- & 20-layer models. The total number of Gaussians is shown in the bottom-right corner.

Table 4: Ablation study on sparsification hyperparameters across different sparse-view settings. We vary the fusion radius ($r$) and the number of sampled fusion centers ($k$). A fusion radius of $r = 0.05$ and the number of sampled fusion centers $k = 30K$ provide the best trade-off between model compactness and fidelity.

| | $r$ | $k$ | Real Dataset | | | | Synthetic Dataset | | | |
| | | | PSNR↑ | SSIM↑ | $N$↓ | Time↓ | PSNR↑ | SSIM↑ | $N$↓ | Time↓ |
|---|---|---|---|---|---|---|---|---|---|---|
| 5 views | 0.03 | 1K | 28.39 | 0.807 | 17K | 5.5m | 25.68 | 0.787 | 31K | 6.8m |
| | 0.03 | 10K | 28.42 | 0.808 | 13K | 5.6m | 25.67 | 0.788 | 26K | 6.6m |
| | 0.03 | 30K | 28.52 | 0.813 | 9K | 5.6m | 25.74 | 0.791 | 19K | 6.5m |
| | 0.05 | 10K | 28.52 | 0.815 | 9K | 5.4m | 25.70 | 0.791 | 18K | 6.1m |
| | 0.05 | 30K | 28.58 | 0.825 | 6K | 5.7m | 25.75 | 0.795 | 11K | 6.0m |
| | 0.10 | 10K | 28.49 | 0.820 | 6K | 5.1m | 25.71 | 0.794 | 11K | 5.6m |
| | 0.10 | 30K | 28.48 | 0.811 | 6K | 5.4m | 25.71 | 0.795 | 8K | 6.2m |
| 10 views | 0.03 | 1K | 32.68 | 0.880 | 22K | 6.1m | 29.30 | 0.852 | 37K | 7.0m |
| | 0.03 | 10K | 32.75 | 0.881 | 17K | 6.2m | 29.32 | 0.853 | 29K | 6.9m |
| | 0.03 | 30K | 32.85 | 0.882 | 12K | 6.5m | 29.37 | 0.855 | 21K | 7.0m |
| | 0.05 | 10K | 32.95 | 0.885 | 11K | 6.2m | 29.34 | 0.855 | 20K | 6.7m |
| | 0.05 | 30K | 33.04 | 0.890 | 9K | 6.8m | 29.45 | 0.858 | 14K | 6.8m |
| | 0.10 | 10K | 33.00 | 0.888 | 9K | 6.2m | 29.40 | 0.856 | 13K | 6.5m |
| | 0.10 | 30K | 32.93 | 0.883 | 8K | 6.2m | 29.43 | 0.857 | 11K | 7.1m |
| 15 views | 0.03 | 1K | 34.30 | 0.900 | 27K | 6.7m | 31.00 | 0.878 | 41K | 7.4m |
| | 0.03 | 10K | 34.38 | 0.900 | 21K | 6.8m | 31.06 | 0.879 | 32K | 7.3m |
| | 0.03 | 30K | 34.50 | 0.901 | 16K | 7.1m | 31.10 | 0.880 | 23K | 7.4m |
| | 0.05 | 10K | 34.52 | 0.902 | 15K | 6.8m | 31.10 | 0.881 | 22K | 7.1m |
| | 0.05 | 30K | 34.62 | 0.903 | 10K | 7.3m | 31.16 | 0.882 | 15K | 7.3m |
| | 0.10 | 10K | 34.52 | 0.902 | 10K | 6.8m | 31.12 | 0.881 | 14K | 6.9m |
| | 0.10 | 30K | 34.50 | 0.902 | 9K | 6.9m | 31.14 | 0.881 | 13K | 7.6m |
| 25 views | 0.03 | 1K | 36.38 | 0.853 | 47K | 9.0m | 33.45 | 0.913 | 42K | 7.9m |
| | 0.03 | 10K | 36.40 | 0.852 | 43K | 9.0m | 33.45 | 0.913 | 36K | 8.0m |
| | 0.03 | 30K | 36.43 | 0.852 | 38K | 9.1m | 33.46 | 0.912 | 31K | 8.1m |
| | 0.05 | 10K | 36.47 | 0.851 | 32K | 8.6m | 33.44 | 0.912 | 27K | 7.7m |
| | 0.05 | 30K | 36.48 | 0.851 | 26K | 8.8m | 33.45 | 0.912 | 21K | 7.9m |
| | 0.10 | 10K | 36.43 | 0.850 | 22K | 8.3m | 33.32 | 0.911 | 18K | 7.5m |
| | 0.10 | 30K | 36.44 | 0.849 | 22K | 9.0m | 33.31 | 0.910 | 17K | 8.1m |

## D.3 MASKING

In Table 5, we present an ablation on the use of the soft Otsu mask during 3D error-guided densification. Particularly in sparse-view configurations, the mask acts as a crucial spatial prior, improving reconstruction quality and model compactness. Without it, the model tends to overfit by placing Gaussians in empty space to minimize 2D projection errors, degrading the 3D geometry and inflating model size. The mask constrains densification to the object's volume, focusing model capacity on refining true geometric details.

Table 5: Ablation on background masking, comparing performance on real and synthetic datasets across different sparse-view settings for a fixed 20-layer model. Mask improves reconstruction quality (PSNR, SSIM) and leads to a more compact model with fewer primitives ($N$).

| | Masking | Real Dataset | | | | Synthetic Dataset | | | |
|---|---|---|---|---|---|---|---|---|---|
| | | PSNR↑ | SSIM↑ | $N$↓ | Time↓ | PSNR↑ | SSIM↑ | $N$↓ | Time↓ |
| 5 views | without | 28.11 | 0.786 | 15K | 5.0m | 25.52 | 0.766 | 20K | 5.8m |
| | with | 28.65 | 0.827 | 6K | 5.6m | 25.73 | 0.794 | 11K | 6.0m |
| 10 views | without | 32.54 | 0.856 | 16K | 5.4m | 29.16 | 0.840 | 20K | 5.8m |
| | with | 33.05 | 0.890 | 9K | 6.7m | 29.43 | 0.858 | 14K | 6.8m |
| 15 views | without | 34.23 | 0.885 | 18K | 5.5m | 30.90 | 0.870 | 20K | 5.8m |
| | with | 34.65 | 0.903 | 10K | 7.0m | 31.14 | 0.882 | 15K | 7.3m |
| 25 views | without | 36.57 | 0.855 | 33K | 6.5m | 33.46 | 0.912 | 24K | 5.7m |
| | with | 36.46 | 0.851 | 26K | 8.7m | 33.39 | 0.913 | 21K | 8.0m |

## D.4 LAYER SELECTION

We investigated several layer selection strategies aimed at reducing computational cost by selectively updating subsets of layers. These included boosting-like approaches, such as training only the newest layer or optimizing a sliding window of recent layers. However, as shown in Table 6, joint optimization of all layers consistently yielded superior results. We attribute this to the need for global coherence: freezing earlier layers prevents them from adapting to the details introduced by new layers.

Table 6: Ablation on layer training strategies for a 20-layer model under the 10-view setting. The superior full-training strategy is highlighted.

| | Strategy | Real Dataset | | | | Synthetic Dataset | | | |
|---|---|---|---|---|---|---|---|---|---|
| | | PSNR↑ | SSIM↑ | $N$↓ | Time↓ | PSNR↑ | SSIM↑ | $N$↓ | Time↓ |
| 10 views | Train newest layer | 32.28 | 0.873 | 20K | 6.9m | 28.98 | 0.841 | 24K | 7.3m |
| | Train last 2 layers | 32.41 | 0.876 | 18K | 6.8m | 29.05 | 0.844 | 23K | 7.5m |
| | Train last 2 layers | 32.41 | 0.876 | 18K | 6.8m | 29.05 | 0.844 | 23K | 7.5m |
| | Train last 3 layers | 32.41 | 0.878 | 17K | 6.7m | 29.04 | 0.844 | 23K | 7.4m |
| | Prob. chain | 32.64 | 0.880 | 13K | 6.4m | 29.25 | 0.851 | 16K | 7.4m |
| | Prob. independent | 32.66 | 0.878 | 13K | 6.3m | 29.19 | 0.850 | 17K | 7.0m |
| | Train all layers | 33.02 | 0.889 | 7K | 6.6m | 29.45 | 0.857 | 14K | 6.8m |

## D.5 SCALING TERM FOR DENSITY INITIALIZATION

In Table 7, we evaluate different scaling functions for density initialization based on the primitive count $N$. By initializing new layers with progressively lower densities, new primitives make gentle corrections to the residual error rather than destabilizing the structure learned by previous layers. We selected $1/\sqrt[3]{N}$ based on our empirical results and the intuition of working in 3D space.

Table 7: Ablation study on the density normalization scaling factor for the 10-view setting. We select $1/\sqrt[3]{N}$ (highlighted).

|  |  | Real Dataset | | Synthetic Dataset | |
| --- | --- | --- | --- | --- | --- |
|  | Scaling | PSNR↑ | SSIM↑ | PSNR↑ | SSIM↑ |
| 10 views | Linear ($1/N$) | 32.65 | 0.871 | 29.33 | 0.854 |
|  | Square Root ($1/\sqrt{N}$) | 33.06 | 0.889 | 29.46 | 0.858 |
|  | Cube Root ($1/\sqrt[3]{N}$) | 33.04 | 0.889 | 29.44 | 0.858 |
|  | None ($1/1$) | 32.94 | 0.889 | 29.19 | 0.854 |

## D.6 CGLS NUMBER OF ITERATIONS

We conducted an ablation study on the number of iterations for CGLS solver in Table Table 8. In sparse-view settings, iterative solvers tend to show semi-convergence: they recover the object's low-frequency structure in the first iterations but eventually begin to overfit high-frequency noise and streaking artifacts. Our results show that reconstruction fidelity peaks at 10 iterations before degrading. We therefore use 10 iterations for residual reconstruction, relying on early stopping as an effective form of regularization.

Table 8: Ablation study on the number of CGLS iterations across different sparse-view settings. We select 10 iterations (highlighted).

|  |  | Real Dataset | | | Synthetic Dataset | | |
| --- | --- | --- | --- | --- | --- | --- | --- |
|  | #iter | PSNR↑ | SSIM↑ | Time↓ | PSNR↑ | SSIM↑ | Time↓ |
| 5 views | 1 | 22.87 | 0.472 | 0.13s | 20.28 | 0.360 | 0.13s |
|  | 3 | 24.52 | 0.582 | 0.25s | 22.59 | 0.547 | 0.24s |
|  | 5 | 24.61 | 0.579 | 0.36s | 22.79 | 0.537 | 0.36s |
|  | 10 | 24.57 | 0.546 | 0.65s | 22.79 | 0.482 | 0.65s |
|  | 20 | 24.57 | 0.546 | 0.65s | 22.79 | 0.482 | 0.66s |
|  | 100 | 24.57 | 0.546 | 0.65s | 22.79 | 0.482 | 0.66s |
| 10 views | 1 | 22.91 | 0.464 | 0.13s | 20.32 | 0.354 | 0.13s |
|  | 3 | 25.65 | 0.630 | 0.26s | 23.78 | 0.603 | 0.26s |
|  | 5 | 26.17 | 0.614 | 0.38s | 24.51 | 0.579 | 0.38s |
|  | 10 | 26.21 | 0.585 | 0.69s | 24.64 | 0.512 | 0.70s |
|  | 20 | 26.08 | 0.553 | 0.89s | 24.48 | 0.467 | 0.90s |
|  | 100 | 26.08 | 0.553 | 0.89s | 24.48 | 0.467 | 0.90s |
| 15 views | 1 | 22.91 | 0.463 | 0.14s | 20.33 | 0.352 | 0.14s |
|  | 3 | 26.04 | 0.655 | 0.27s | 24.19 | 0.629 | 0.27s |
|  | 5 | 26.92 | 0.637 | 0.40s | 25.26 | 0.601 | 0.40s |
|  | 10 | 27.18 | 0.611 | 0.73s | 25.61 | 0.535 | 0.74s |
|  | 20 | 26.69 | 0.518 | 1.24s | 24.99 | 0.406 | 1.27s |
|  | 100 | 26.69 | 0.518 | 1.24s | 24.99 | 0.406 | 1.26s |
| 25 views | 1 | 22.93 | 0.482 | 0.17s | 20.35 | 0.356 | 0.16s |
|  | 3 | 26.67 | 0.703 | 0.33s | 24.59 | 0.679 | 0.31s |
|  | 5 | 28.08 | 0.689 | 0.50s | 26.61 | 0.677 | 0.46s |
|  | 10 | 28.25 | 0.673 | 0.92s | 27.99 | 0.664 | 0.84s |
|  | 20 | 26.89 | 0.633 | 1.66s | 28.26 | 0.633 | 1.61s |
|  | 100 | 25.62 | 0.561 | 2.18s | 28.16 | 0.601 | 2.08s |

## D.7 3D GAUSSIAN BLUR

In Table 9, we present an ablation on the Gaussian blur parameter ($\sigma$) applied to the 3D error map prior to sampling. This step is used for suppressing high-frequency noise inherent in sparse-view tomographic reconstruction. The results demonstrate a trade-off: lower values fail to filter noise

streaks, while excessively high values may oversmooth the error signal. We find that $\sigma = 2.0$ provides the optimal balance.

Table 9: Ablation study on the Gaussian blur parameter ($\sigma$) applied to the 3D error map for the 5-view setting. We vary $\sigma$ to control the suppression of high-frequency noise before sampling. We find $\sigma = 2.0$ (highlighted) provides the best balance.

|  | $\sigma$ | Real Dataset | | Synthetic Dataset | |
|---|---|---|---|---|---|
|  |  | PSNR↑ | SSIM↑ | PSNR↑ | SSIM↑ |
| 5 views | 0.1 | 28.45 | 0.826 | 25.67 | 0.793 |
|  | 0.5 | 28.44 | 0.830 | 25.68 | 0.792 |
|  | 1.0 | 28.57 | 0.829 | 25.74 | 0.793 |
|  | 2.0 | 28.62 | 0.829 | 25.73 | 0.794 |
|  | 10.0 | 28.60 | 0.824 | 25.72 | 0.790 |

## D.8    GUMBEL SAMPLING TEMPERATURE

In Table 10, we evaluate the impact of the temperature parameter ($\tau$) used in Gumbel-Max sampling strategy. This parameter controls the entropy of the sampling distribution derived from the 3D error map. A near-zero temperature approaches a deterministic argmax operation. In sparse-view tomography, these maxima often correspond to noise spikes or streak intersections rather than true missing geometry. Conversely, excessively high temperatures flatten the distribution, causing the model to ignore the error guidance and sample uniformly. Our results show that $\tau = 0.005$ offers the best robustness, allowing the model to sample broadly from the high-error regions to recover structure while avoiding overfitting to specific high-frequency noise artifacts.

Table 10: Ablation study on the Gumbel-Max sampling temperature ($\tau$) for the 5-view setting, where sensitivity to noise is most critical. We select $\tau = 0.005$ (highlighted) to balance diverse sampling from high-error regions with robustness against noise artifacts.

|  | $\tau$ | Real Dataset | | Synthetic Dataset | |
|---|---|---|---|---|---|
|  |  | PSNR↑ | SSIM↑ | PSNR↑ | SSIM↑ |
| 5 views | 0.0005 | 28.49 | 0.817 | 25.75 | 0.793 |
|  | 0.001 | 28.46 | 0.819 | 25.76 | 0.792 |
|  | 0.005 | 28.64 | 0.830 | 25.72 | 0.793 |
|  | 0.01 | 28.58 | 0.824 | 25.68 | 0.794 |
|  | 0.05 | 28.44 | 0.817 | 25.63 | 0.793 |

## D.9    TV LOSS

In Table 11, we examine the impact of the Total Variation regularization weight ($\lambda_{\text{TV}}$). This loss functions as a general smoothness prior by penalizing local gradient magnitudes across the entire volume. Because the penalty is applied uniformly to all spatial gradients, it suppresses noise but may also suppress real structural details. The results reflect this trade-off: lower weights fail to contain sparse-view noise, while higher weights lead to an over-smoothed representation. We adopt $\lambda_{\text{TV}} = 0.25$ to maintain a baseline of structural coherence.

Table 11: Ablation study on the Total Variation regularization weight ($\lambda_{\text{TV}}$). We find $\lambda_{\text{TV}} = 0.25$ (highlighted) provides the optimal balance, maximizing reconstruction quality.

| | $\lambda_{\text{TV}}$ | Real Dataset | | Synthetic Dataset | |
| --- | --- | --- | --- | --- | --- |
| | | PSNR↑ | SSIM↑ | PSNR↑ | SSIM↑ |
| 5 views | 0.05 | 28.20 | 0.811 | 24.80 | 0.751 |
| | 0.25 | 28.75 | 0.828 | 25.67 | 0.788 |
| | 0.50 | 28.56 | 0.829 | 25.75 | 0.793 |
| | 0.75 | 28.44 | 0.823 | 25.71 | 0.794 |
| 10 views | 0.05 | 33.39 | 0.884 | 28.73 | 0.833 |
| | 0.25 | 33.59 | 0.892 | 29.65 | 0.858 |
| | 0.50 | 32.99 | 0.888 | 29.41 | 0.857 |
| | 0.75 | 32.52 | 0.883 | 29.15 | 0.854 |
| 15 views | 0.05 | 35.72 | 0.906 | 31.25 | 0.875 |
| | 0.25 | 35.47 | 0.908 | 31.62 | 0.886 |
| | 0.50 | 34.59 | 0.903 | 31.13 | 0.882 |
| | 0.75 | 33.84 | 0.897 | 30.69 | 0.876 |
| 25 views | 0.05 | 35.29 | 0.840 | 35.03 | 0.925 |
| | 0.25 | 36.46 | 0.850 | 34.43 | 0.922 |
| | 0.50 | 36.41 | 0.851 | 33.46 | 0.912 |
| | 0.75 | 36.00 | 0.849 | 32.71 | 0.904 |

