# OpenReview forum: "Layer-Based 3D Gaussian Splatting for Sparse-View CT Reconstruction"
_ICLR.cc/2026/Conference — Submitted to ICLR 2026_

### Official Review · Reviewer_qVn8 · 2025-10-26

**Soundness:** 3
**Presentation:** 3
**Contribution:** 3
**Rating:** 6
**Confidence:** 5

**Summary:**

This paper introduce a hierarchical layer-based 3D Gaussian Splatting (3DGS) computed tomography reconstruction framework. The reconstructed objects are iteratively refined by correcting the volumetric errors of previous layers. The core technical contribution is the 3D error-driven strategy to guide densification and sparsification. This strategy estimates a volumetric error map from back-projected 2D residuals, providing direct structural guidance for adding Gaussians in underrepresented regions and fusing them in over-represented regions.

**Strengths:**

(i) The idea of hierarchical layer-based 3D Gaussian Splatting is novel and interesting and insighted as the basic shape of the scanned object can be easily reconstructed while the fine-grained details are hard to captured. Existing 3DGS-based method usually neglect this while this work fills this research gap. The designed technique, 3D error-guided importance sampling is also very reasonable by adding Gaussians in the underrepresented regions estimated by the positive error maps and fusing them in the over-represented regions estimated by the negative error maps.

(ii) The performance on 3D CT reconstruction is solid. This work is based on the NeurIPS 2024 work R2-Gaussian. By applying the new densification and sparsification strategy to the baseline method, the performance are improved significantly by large margins on both real and synthetic datasets, as shown in Table 1. These results suggest the effectiveness of the proposed method. The visual comparison in figure can also show the propose method reconstructs clearer structural details.

(iii) The overall writing is clear, especially the method part from line 187 to line 288. The presentation is also well-dressed. The workflow of the pipeline is clearly shown in the figure 2. I like the style as almost all the technical contributions are reflected in the figure.

(iv) The ablation study is pretty comprehensive. All the modification are investigated, including the layered densification, layer sparsification, Masking, layer selection, and so on. The results in Table 2, 3, 4, 5, and 6 can clearly demonstrate the effectiveness of the proposed technical modifications.

(v) Code has been submitted. The reproducibility can be checked.

**Weaknesses:**

(i) The motivation is not very clear. As described in Line 038 – 042, why the regular 3DGS provides only indirect and incomplete information about the true 3D structure is not well discussed. From my point of view, this paper mainly modifies the densification of the Gaussian point clouds and the initialization has not been improved. So why also mention the one-time initialization here? It is a little weird.

(ii) The differences of the proposed densification and sparsification strategies and the regular ones should be highlighted and comprehensively compared. Now the authors just plainly describe their methods. I suggest the author draw some figures and mention the differences in the method section. Meanwhile, in the teaser figure, the authors just show the changes of the Gaussian point clouds of the proposed method. How about the regular one? There is no more comparison to show the advantages of the proposed method.

(iii) The main results are not very convincing. The authors claim their method beats the state-of-the-art methods but they do not use the public benchmark – X3D and did not make comparisons to the recent best neural radiance fields (NeRFs) method – SAX-NeRF, which was published by CVPR 2024. Instead, they compare with an old method NAF, which was accepted by MICCAI 2022.

(iv) The main visual results also look very weird because the color is somewhat red, which is significantly different from the visual results shown in previous works such as NAF, SAX-NeRF, X-Gaussian, R2-Gaussian, etc. There is no explanation for this.

**Questions:**

(i) Could you please explain why using the TV loss in Eq.(6)? Do you do an ablatio study of this loss function?

---

> ### Author Response · Authors · 2025-11-21
>
> We thank the reviewer for their comments. We have addressed them and we discuss them in the following.
>
> **A1. Motivation of paper:** Thank you for pointing it out, we agree that we have not discussed this point properly. We have rewritten the introduction to make it clearer. The signal used by standard 3DGS densification (gradients from 2D projection errors) might be indirect (a 2D error does not uniquely specify a 3D location) and incomplete (due to missing viewing angles). Our method addresses this by exploiting the known scanner geometry to reconstruct a direct 3D error map from all available views. The reviewer is correct that our method modifies the densification process. However, instead of standard local, gradient-driven densification, we employ a global, error-guided, iterative *initialization* process. This is again different from standard 3DGS which initializes the representation once and then locally refines it via splitting/cloning. Our method iteratively constructs the scene by initializing entirely new layers of primitives from scratch, guided by the global 3D error map. Therefore, we mention "one-time initialization" to contrast the standard paradigm (initialize once, then locally refine) with our paradigm (iteratively initialize new layers to globally refine). Thus, our "densification" is a series of error-guided *initializations*.
>
> **A2. Differences in sparsification and densification:** We will include a proper visualization to highlight the differences between our densification/sparsification method and the classic ones. In the following, we discuss them in more detail.
> Traditional 3DGS densification (Kerbl et al., 2023) operates locally: Gaussians with high accumulated 2D gradients are split or cloned. While effective for dense-view scenarios, this strategy has two limitations in sparse-view CT: (i) indirect 3D information – a 2D gradient indicates where in the image refinement is needed, but not where in 3D space the error originates; (ii) view-dependent adaptation – densification responds to each view independently, potentially overfitting to observed angles while leaving unobserved regions under-represented. Our approach addresses both issues: we reconstruct an explicit 3D error map from all views (providing direct 3D guidance), and add structured layers globally (ensuring multi-view consistency).
>
> **A3. Additional comparisons:** While our work primarily focuses on explicit representations, for completeness, we can add SAX-NeRF comparisons to the final version. Regarding the dataset, there is a significant overlap in scenes with the X3D dataset. We can also add results on the few missing non-overlapping scenes to the final version of the paper.
>
> **A4. Color of visualization:** Thank you for pointing this out. We have updated visualizations to use a gray color for consistency with prior works.
>
> **Q1. TV loss:** We use the 3D Total Variation (TV) loss as a regularization term. This loss functions as a general smoothness prior by penalizing local gradient magnitudes across the entire volume. Since it applies smoothing everywhere, it helps remove noise, but a heavy regularization can blur out real details. We performed an ablation study (*cf.* $\S$ D.9) which demonstrates this trade-off. Increasing the TV regularization weight $\lambda_{\text{TV}}$ helps with artifacts common in sparse-view reconstruction. However, an aggressive TV weight can lead to over-smoothing, causing a loss of fine details. Our final choice of $\lambda_\text{TV}$ for our method was selected to balance these two effects. Based on this analysis, we refined our parameter choice from 0.5 to 0.25 (cf. Table 11), which yields the best balance. We updated accordingly the main table (*cf.* Table 1) and per-scene comparison (cf. Table 2). Regardless of the TV weight, our model consistently outperforms the baseline.

---

### Official Review · Reviewer_1Nja · 2025-10-30

**Soundness:** 3
**Presentation:** 2
**Contribution:** 2
**Rating:** 6
**Confidence:** 4

**Summary:**

This paper proposed a hierarchical approach to 3D gaussian splatting for sparse-view CT reconstruction, first introducing large-scale Gaussians, and then refining in later steps. Refinement choices are based on reconstruction of the residual error.

**Strengths:**

- The paper addresses an important challenge in CT, and the proposed method seems reasonable for the challenge.

- Experimental results are  included for both synthetic and real-world datasets.

**Weaknesses:**

- The main contributions of the paper are not clearly described. Both Gaussian splatting and hierarchical reconstruction approaches are quite well established in the CT community (as the authors state), so it is important to clearly state how the proposed paper exactly contributes to the already known knowledge.

- It is not clear to me why a negative reconstructed error indicates areas where overfitting or redundancy is present. For example, if a sample has a small hole somewhere, and the algorithm represents that part with a single large Gaussian, a negative error would occur in the error map, but the model is actually underfitting. The specific choice of interpreting negative as overfitting and positive as underfitting should be clearly motivated in the paper, and ideally evidence should be given that this is indeed a valid choice.

- It is not clear how specific hyperparameters settings are chosen by the authors, and how results are affected by different choices for the hyperparameters. This is also try for comparison methods -- how are the hyperparameters chosen for these?

- The terminology used, especially the use of 'layer' is unclear. 'Layer' has a very specific meaning in deep learning, so using it for a completely different concept is confusing. I suggest using a different term.

**Questions:**

- What are the main contributions of the paper?

- Why does a negative error indicates overfitting?

- How were hyperparameters chosen, and how do hyperparameters affect results?

---

> ### Author Response · Authors · 2025-11-21
>
> We thank the reviewer for their comments. We have addressed them and we discuss them in the following.
>
> **A1. Main contributions not clearly described:** We have rewritten some parts of the introduction to highlight better our methodological contributions and the differences with the previous approaches to the problem. In particular, a key novelty lies in how and where new primitives are introduced. Standard Gaussian splatting densification is a local process driven by gradient thresholds (splitting/cloning) based on 2D supervision. This approach is spatially constrained, as *new primitives can only appear in the immediate vicinity of their parents*, and it is prone to view-dependent overfitting. Our method, in contrast, entails a global construction process. We use a 3D volumetric error map to guide the initialization of new Gaussian layers. This allows us to place new primitives anywhere in the 3D volume where there is a structural error. This is an iterative residual fitting in 3D, where each new layer is explicitly tasked with modeling the volumetric errors left by the previous ones. At the same time, it is fundamentally different from other hierarchical methods. Please refer to Rev. Cwpe A2 for further details.
>
> **A2. Negative error = overfitting?:** Thank you for raising this issue and the stimulating example. We agree that the use of *overfitting* and *underfitting* might be misleading. In general, removing Gaussians (as in the case of negative error) might increase the bias of the model leading to underfitting, while including new Gaussians might increase the variance leading to overfitting. Since the influence of the number of Gaussians on the bias-variance tradeoff is not trivial, we have changed the phrasing in the paper and used more neutral terms such as *over-representation* and *under-representation*. The hole example is particularly interesting since it can describe an over-representation due to the presence of a large Gaussian and a following under-representation due to a *potential* lack of smaller Gaussian modeling its borders, which will then be addressed by our model in a successive step, when a positive error is detected.
>
> **A3. Choice of the hyperparameters settings:** For the classical baselines we tuned the number of iterations and chose the value that yielded the best average performance (e.g., 10 iterations for CGLS, 20 for SART, and 100 for SART-TV and ASD-POCS). For R$^2$ Gaussian, we set up the parameters to create a stronger baseline specifically for the sparse-view setting, as the original settings may produce artifacts. This process is detailed in Appendix C and Table 2. For our method, we performed ablation studies to determine a robust set of hyperparameters. Importantly, we used a single set of parameters for all experiments across all datasets and view counts to demonstrate the generalizability of our approach, rather than overfitting by tuning hyperparameters individually for each specific case.
>
> **A4. ”Layer” term:** We agree that the term "layer" has a specific meaning in the context of neural networks. Our use of the term is instead borrowed from the fields of computer graphics, digital imaging, and hierarchical representations. In these fields “layer" refers to a distinct set of components that are composited together to form a final result. In our case, the term ”layer” refers to a new set of Gaussians added at a specific stage to address remaining volumetric errors. We have included a footnote in the paper for better clarification.

---

> > ### Comment · Reviewer_1Nja · 2025-11-27
> >
> > I want to thank the authors for their response. Although the authors addressed some of my comments, my main concerns were not taken away. For example, there is still not a lot of explicit evidence for the validity of the choice of method to add or merge gaussians. Also, the specific choices for comparison method parameters is still not clear (what was the TV strength for the TV methods? How was this chosen? Was a nonnegativity constraint tried, and if not, why not? etcetera). Therefore, I will keep my score as 6: marginally above the acceptance threshold. But would not mind if paper is rejected.

---

> > > ### Author Response · Authors · 2025-12-02
> > >
> > > **Evidence for the validity of the addition/merging of Gaussians.** Positive values in the 3D error map indicate under-representation, guiding the insertion of new Gaussians at the spatial locations with the highest positive error to explicitly recover missing structure. Conversely, negative values indicate over-representation, guiding the merging of Gaussians to reduce redundancy. This logic is empirically shown in *Table 3*, where our error-guided addition outperforms the single-layer baseline where such guidance is absent, and *Table 4*, where this merging strategy improves both fidelity and compactness. We also note that Reviewers WV1U (S1), Cwpe (S1), and qVn8 (S2) have acknowledged the soundness of this error-guided allocation strategy.
> > >
> > > **Baseline Configuration (Constraints and TV strength).** We adhered to standard implementations in the TIGRE toolbox (Biguri et al., 2016).
> > > - Non-negativity: E.g., for SART, SART-TV, we enforced the default non-negativity constraint. Disabling the constraint (noneg=False) consistently degrades performance, as it ignores the physical non-negativity of X-ray attenuation. For example, with SART on 5-view real scenes, the average PSNR drops from 25.84 dB (default) to 24.69 dB when the constraint is removed.
> > > - TV strength: We utilized default parameters for regularized methods (e.g.,  $\lambda = 50$ for SART-TV, $\alpha = 0.002$ for ASD-POCS). Although increasing the TV weight may produce slight improvements, it is not sufficient to match the performance of our method. For example, with ASD-POCS on 5-view real scenes, the average PSNR marginally increases from 26.64 dB (default) to a peak of 26.77 dB.
> > >
> > > **CGLS and residual reconstruction.** In contrast, standard CGLS is *unconstrained by design*. We leveraged this in our framework: since our guiding 3D error map represents residuals, it must capture both positive values (under-representation → error-guided densification) and negative values (over-representation → error-guided sparsification). While other unconstrained solvers (e.g., with regularization) could be employed, we selected standard CGLS *for its computational efficiency* in the iterative training loop, managing noise via a lightweight post-process instead.

---

### Official Review · Reviewer_Cwpe · 2025-10-31

**Soundness:** 2
**Presentation:** 2
**Contribution:** 2
**Rating:** 4
**Confidence:** 3

**Summary:**

This paper proposes a hierarchical, layer-based 3D Gaussian Splatting (3DGS) framework for sparse-view CT. Instead of a one-shot dense initialization, the method adds new layers of smaller Gaussians in a coarse-to-fine manner, where placement is guided by a 3D volumetric error map reconstructed via back-projecting 2D residuals with CGLS. Positive-error regions trigger densification (adding Gaussians); negative-error regions trigger sparsification (fusing Gaussians). The system starts from a classical reconstruction (SART-TV) to derive a soft Otsu mask and to seed the first layer. Experiments on synthetic and real datasets show improved 3D PSNR/SSIM over traditional solvers and prior explicit/implicit baselines (notably R$^2$-Gaussian), especially at very sparse views (5–15).

**Strengths:**

1. The layerwise residual-fitting idea is well-motivated and implemented end-to-end in CT with explicit 3D error maps driving where capacity is allocated.
2. Experiment in Table 1 shows improvements vs. strong baselines at 5–15 views on both real and synthetic sets (e.g., Real/10 views: PSNR 33.04 vs. 31.90 for R$^2$-Gaussian). Qualitative figures support crisper geometry with fewer view artifacts.
3. The paper varies number of layers, sparsification radius/centers, masking, and layer-optimization strategies; the 20-layer configuration emerges as the best trade-off and uses fewer final Gaussians with competitive time.

**Weaknesses:**

1. The R$^2$-Gaussian has been missed spelled as R2-Gaussian in all places in this paper, which is very non-professional.
2. The core idea of hierarchically allocating Gaussian capacity is closely parallels existing hierarchical/level-of-detail 3DGS schemes and explicit voxel/atom allocation strategies. The paper does not clearly articulate a substantive technical advance beyond adapting these known capacity-allocation ideas to CT, nor does it provide head-to-head analyses that convincingly demarcate what is genuinely new.
3. The method critically depends on the CGLS-reconstructed 3D error map. The discussion admits noise/streaks in highly sparse regimes and uses mask+Gaussian blur to denoise, but quantitative sensitivity to solver iterations, regularization, and noise level is limited.
4. The paper regularizes R$^2$-Gaussian for sparse views (higher TV, minimum scale, densification threshold). While this avoids needle artifacts, it may underplay R$^2$-Gaussian’s potential at moderate views and shifts the comparison space.

**Questions:**

Please refer to the weaknesses part, especially weakness 2&3. I would like to read the rebuttal and improve my ratings if my concerns are adequately addressed.

---

> ### Author Response · Authors · 2025-11-21
>
> We thank the reviewer for their comments. We have addressed them and we discuss them in the following.
>
> **A1. $R^2$-Gaussian name:** Thank you for pointing out this issue. We have corrected all occurrences in the revised version to match the official paper.
>
> **A2. Differences with other methods:** In the following, we discuss the main differences of our method with the existing work. We have included it in the main paper (Introduction + Related work sections).
>
> Existing hierarchical methods (Kerbl et al., 2024; Müller et al., 2022; Zha et al., 2022; Rückert et al., 2022),  organize primitives by dividing the volume into independent chunks (octree nodes, hash grid cells) for efficiency. Primitives are *locked* to their assigned chunk and optimized *independently*. This approach might be suboptimal in sparse-view CT for several reasons: (i) in sparse-view settings, many local cells have sparse ground truth signal, leading to inconsistent solutions at cell boundaries; (ii) such structures typically rely on fixed subdivision rules. If the initial subdivision is incorrect due to sparsity, the method cannot easily recover.
>
> While our method is hierarchical, its core mechanism is fundamentally different from existing methods. The fundamental distinction lies in the implementation of a non-rigid *hierarchy induced by residuals*, rather than a spatial one. In this sense, our approach is *holistic*, in that we optimize the representation *globally* rather than partitioning it into independent, localized units. As a result, all layers coexist in the *same 3D space* and are *jointly optimized* to ensure global consistency.
>
> Additionally, standard 3DGS densification relies on accumulated 2D projection gradients to decide where to split/clone Gaussians. This signal is indirect: a 2D gradient indicates refinement is needed but does not specify the 3D location. Instead, our method exploits CT's known *scanner geometry* to aggregate 2D residuals from all views and reconstruct an explicit volumetric error map for 3D guidance.
>
> Finally, differently from prior work, we not only perform densification but also implement a *sparsification* strategy in over-represented regions (negative error), for further regularization of the model.
>
> **A3. Sensitivity to CGLS Algorithm:** Thank you for pointing it out. To test the sensitivity to the CGLS algorithm, we have performed an ablation study of the hyperparameters you have mentioned, choosing the configurations that reduce noise in the 3D error-map reconstruction process. We provide the results in Appendix $\S$ D6, D3, D7, D8 (Tables 8, 5, 9, 10). In particular, we found that:
>
> - Reconstruction fidelity peaks at 10 iterations before degrading. This occurs because many iterative solvers, including CGLS, recover the low-frequency structure of the signal during the initial iterations and eventually begin overfitting to high-frequency noise and streaking artifacts. We therefore use 10 iterations for the residuals, relying on early stopping as an effective form of regularization.
> - As expected, masking, 3D Gaussian blur, and probabilistic sampling prevent overfitting to background noise, due to suppression of high-frequency artifacts. Specifically, masking removes background noise, blur smooths out reconstruction streaks, and probabilistic sampling prevents the model from deterministically overfitting to isolated noise peaks.
>
> *Cf.* Rev. WV1U A1 for more detail.
>
> **A4. Regularization of $R^2$-Gaussian model:** Thank you for pointing this out. As detailed in Appendix C and Table 2 in the paper, we ensured a more fair comparison by adjusting the R$^2$-Gaussian parameters within a reasonable range in favor of the sparse-view setting. In fact, when trained with its original default parameters, R$^2$-Gaussian may overfit under sparse-view conditions, producing needle-like artifacts. To provide a stronger baseline, we set up regularization parameters specifically to improve its performance in sparse-view regimes. This introduces an inherent trade-off, since a less-regularized model performs better at higher view counts (e.g., 25 views), whereas a more-regularized configuration suppresses artifacts but tends to oversmooth. Even with these favorable adjustments, our layer-based strategy outperforms R$^2$-Gaussian in sparse-view settings.

---

### Official Review · Reviewer_WV1U · 2025-11-01

**Soundness:** 3
**Presentation:** 3
**Contribution:** 3
**Rating:** 6
**Confidence:** 5

**Summary:**

This work introduces a hierarchical, layer-based framework for 3D Gaussian Splatting (3DGS) tailored for sparse-view CT reconstruction. The core contribution is a 3D error-guided refinement strategy, where 2D projection residuals are back-projected using a tomographic solver to create a 3D volumetric error map. This 3D error map directly guides a coarse-to-fine process, acting as an adaptive importance sampling mechanism for both adding new, smaller Gaussians (densification) in under-represented regions (positive error). The error map also guides the merging of existing Gaussians (sparsification) in over-represented regions (negative error), effectively regularizing the model against overfitting.

**Strengths:**

1. The method directly addresses a key failure mode of standard 3DGS in sparse-view settings: overfitting to 2D projections. Guiding densification and sparsification with a 3D error map (from back-projected 2D residuals) is a novel and more geometrically sound approach than relying on 2D gradient-based density control alone.

2. The proposed layered, coarse-to-fine refinement strategy is well-motivated and empirically effective. Ablation studies (Table 3) clearly show that this layered approach outperforms a single-stage, dense initialization, while often converging to a more compact model (fewer Gaussians) and reducing training time.

3. The paper demonstrates state-of-the-art results, consistently outperforming strong baselines (including classical methods, implicit fields like NAF, and 3DGS methods like R2-Gaussian) on both synthetic and real-world datasets, especially in highly sparse (5-15 view) scenarios.

**Weaknesses:**

1. The quality of the 3D error map, which is central to the method, is dependent on the CGLS solver and the quality of the 2D residuals. As acknowledged by the authors, this map can become noisy in extremely sparse settings, potentially leading to error amplification where noise is densified. While denoising is applied, the robustness of this feedback loop could be analyzed further.

2. The normalization term for initializing new Gaussian density, $\alpha_{i}^{(l)}=C_{\alpha}\frac{e_{i}^{(l)}}{\sqrt[3]{N^{(l-1)}}}$ (Equation 4), is heuristic. It is "motivated by the physical process" but relies on a "quasi-uniform distribution" assumption. A more principled derivation or analysis of this scaling factor would strengthen the method's technical foundation.

3. The layer-building process seems to be on a fixed schedule (2500 Gaussians every 500 iterations for 20 layers). An adaptive strategy, where the number of new Gaussians and the timing of new layers are determined by the 3D error map's magnitude or distribution, would be a more elegant and efficient extension (as noted in the discussion).

**Questions:**

Please refer to the weakness part.

---

> ### Author Response · Authors · 2025-11-21
>
> We thank the reviewer for their comments. We have addressed them and we discuss them in the following.
>
> **A1. Quality of the error map:** The CGLS solver serves exclusively as a guidance signal for the initialization of a new layer. Empirically, the quality of the CGLS solver does not limit the final reconstruction quality (*cf.* Table 1 for CGLS vs. Our method). It is true that noisy reconstructions of error can arise, especially in *later iterations* (as discussed in the paper). *Their impact is, however, limited since the global structure of the representation is already established*. Additionally, we have also performed an ablation over the number of CGLS iterations and denoising hyperparameters to assess their effect on the reconstruction quality and included them in Appendix $\S$ D6, D3, D7, D8 (Tables 8, 5, 9, 10). Our ablation studies confirm that *each hyperparameter functions effectively to filter artifacts*, leading to robust performance across all datasets. Please refer also to Rev. Cwpe A3 for further details.
>
> **A2. Normalization term in Gaussian initialization:**  Thank you for pointing this out. The choice of the normalization constant is loosely inspired by the 3D dimensionality of the problem, and we agree with the reviewer that it is a heuristic. However, as empirically shown in our ablation study (*cf.* Section D.5), by initializing new layers with progressively lower densities (scaled by the current capacity of the model), new primitives make gentle corrections to the residual error rather than destabilizing the structure learned by previous layers. We have revised Section $\S$ 3.2.
>
> **A3. Number of Gaussians in layer building process:** We have chosen a fixed number of Gaussians to show the robustness and effectiveness of our method, even without complex adaptive schedules. We agree with the reviewer that an adaptive strategy can further improve the performance of our method, however, designing such a solution is non-trivial and most likely introduces additional hyperparameters, which further complicate the tuning of our method. As future work, we plan to theoretically derive the number of Gaussians using approximation theory, based on the error distribution.

---

### Author Response · Authors · 2025-12-02

We thank all reviewers for their constructive feedback and appreciate that they recognized our method as *"well-motivated"* [WV1U, Cwpe], *"novel"* [qVn8, WV1U], and *"reasonable"* [1Nja, qVn8].

Below we summarize the main changes to the paper:
- We clarified our main contributions *(lines 088 - 098)*, highlighted the novel aspects of our model and their motivations *(lines 037 - 044)*, as well as distinctions from existing hierarchical work *(lines 068 - 078)* [Rev. WV1U, Cwpe, 1Nja, qVn8].
- We clarified the naming of key concepts *(“layer”: lines 045 - 047; “under-represented” and “over-represented”: lines 051 - 067)* [Rev. 1Nja].
- We clarified the selection of hyperparameters via additional ablations, including the number of CGLS iterations, TV regularization, and others *(D5 - D9)* [WV1U, Cwpe, 1Nja]. Based on this analysis, we changed TV weight from 0.5 to 0.25, which led to a slight improvement in the results, and we updated Tables 1 and 2 [Rev. qVn8].
- We corrected typos, and performed stylistic changes in the figures [Rev. Cwpe, qVn8].

---

### Meta-Review · Area_Chair_c93w · 2026-01-07

**Summary:**

The submission proposes a coarse-to-fine 3D Gaussian Splatting approach to CT reconstruction.  The method refines Gaussian primitives in a layerwise fashion.  Improvements of <1db from baselines of 29.94 up to 35.52 PSNR, as well as marginal improvements in SSIM are reported depending on the number of sparse views.

**Reviewer Concerns:**

Main concerns from reviewers include: novelty and motivation of the approach over other hierarchical methods, baseline comparisons, and qualitative appearance of examples.  The rebuttal primarily promises future comparisons, while asserting novelty.

**Reviewer Scores:**

Reviewer scores were 3 reviewers with 6s, and one reviewer who scored a 4.  It is not clear that there would be much movement in those scores on the basis of the rebuttal.  On the balance, there remain concerns about novelty, motivation, and empirical results.

---

### Decision · Program_Chairs · 2026-01-26

Reject